# Effectiveness of school-based interventions in delaying sexual debut among adolescents in sub-Saharan Africa: a protocol for a systematic review and meta-analysis

Beatrice W Maina  ,[1,2] Kenneth Juma  ,[1] Emmy Kageha Igonya,[1] Jane Osindo,[3] Hesborn Wao,[3] Caroline W Kabiru[3]

► Prepublication history and additional online supplemental material for this paper are available online. To view these files, please visit the journal online (http://dx.doi.org/10.1136/bmjopen-2020-044398).

[1]Population Dynamics and Reproductive Health, African Population and Health Research Center, Nairobi, Kenya
[2]School of Public Health, University of the Witwatersrand Faculty of Health Sciences, Johannesburg, South Africa
[3]Research Capacity Strengthening, African Population and Health Research Center, Nairobi, Kenya

**Correspondence to**
Dr Beatrice W Maina;
bmaina@aphrc.org

## ABSTRACT

**Introduction** Early sexual debut is associated with poor sexual and reproductive health outcomes across the life course. A majority of interventions aimed at delaying sexual debut among adolescents in sub-Saharan Africa (SSA) have been implemented in schools with mixed findings on the effectiveness of such interventions. This systematic review will summarise and synthesise existing evidence on the effectiveness of school-based interventions in delaying sexual debut among adolescents aged 10–19 years.

**Methods and analysis** We will conduct a comprehensive database search of peer-reviewed studies published in PubMed, Scopus, Science Direct, Web of Science, HINARI and EBSCO (PsycINFO, Global Health, CINAHL) and in Cochrane library, National Institute of Health and Turning Research into Practice databases for ongoing studies yet to be published. All studies conducted in SSA between January 2009 and December 2020, regardless of the study design, will be included. Two authors will independently screen all retrieved records and relevant data on sexual debut extracted.

Data will be pooled using the random effects model. Dichotomous outcomes will be reported as risk ratios and continuous data as mean difference at 95% CI. Heterogeneity will be assessed using the I² statistic. Findings will be presented in tables and charts, while providing a description of all included studies, themes and concepts drawn from literature.

**Ethics and dissemination** Ethical approval is not required. The findings will be disseminated through peer-reviewed publications, presentations at relevant conferences and other convening focusing on adolescent sexual and reproductive health.

## Strengths and limitations of this study

► To the best of our knowledge, this is the first systematic review and meta-analysis to focus on the effectiveness of school-based interventions on delaying sexual debut among adolescents in sub-Saharan Africa.

► The systematic review and meta-analysis includes all interventions with a school-based component regardless of the study design used in the study.

► The systematic review and meta-analysis will adhere to the Preferred Reporting Items for Systematic Reviews and Meta Analyses guidelines.

► To minimise the likelihood of reviewer bias, two reviewers will screen for study eligibility and perform the quality assessment.

► This review will be limited to peer-reviewed studies, published in English between January 2009 and December 2020

## BACKGROUND

Adolescents aged 10–19 years in sub-Saharan Africa's (SSA) have a high prevalence of risky sexual behaviours including early sexual debut and unsafe sexual practices.[1 2] Consequently, they are most-at-risk for poor sexual and reproductive health (SRH) outcomes.[3–5]

Early sexual debut, described as having had the first sexual intercourse at or before the age of 14 years, increases the period in which an adolescent girl is at risk of getting pregnant[6 7] and is predominantly driven by individual, familial, contextual and sociocultural factors.[8–10] Early sexual debut is also associated with occurrence of sexual violence, unsafe abortions, unplanned pregnancies, early child marriages, sexually transmitted infections and HIV infection,[11–13] elevated risk of cervical cancer[14 15] and poor schooling outcomes.[11 16 17]

Implications of early sexual debut disproportionately affect women and girls, impacting on their health, socioeconomic lives and overall well-being across the life course.[18–26] For instance, young girls who get pregnant at an early age are likely to drop out of school increasing their risk of poor educational and

other socioeconomic outcomes.[27 28] Young girls are also at greater risk of poor maternal outcomes compared with older women. A review of maternal mortality among adolescents compared with women of other ages in 144 countries revealed a threefold higher mortality risk among adolescent mothers compared with women above 30 years.[2] Similarly, adolescents have higher risk of adverse perinatal outcomes, including low birth weight, preterm delivery and perinatal death.[29–31] The strong association between poor maternal education and poor child–health outcomes such as experiencing severe acute malnutrition, infections as well as poor cognitive growth,[32–34] implies that the consequences of early sexual debut transcends generations. Delaying sexual debut is, therefore, a key strategy in averting poor SRH outcomes during and after adolescence.

School is one institution where most adolescents spend most of their time, learning and interacting with peers and adults. Critical thinking developed in schools can be useful for questioning unhealthy behaviours. Schools thus offer a platform for socialisation into healthy and unhealthy behaviours.[35] As attention turns towards meeting the Sustainable Development Goals, the global health community is increasingly committed to adolescent SRH as a prerequisite for improving lives and health of young people.[36] This commitment is exemplified by the vast number of diverse school-based interventions targeted at promoting adolescent SRH across the globe, with limited but growing interest in SSA. For instance, schools are being targeted as sites for provision of age-appropriate comprehensive sexuality education that facilitates improved self-efficacy, knowledge and life skills.[37–39] Other related school-based interventions such as school fees waivers, supply of menstrual products for girls, school feeding programmes for vulnerable populations are also being implemented.[40 41]

However, impact evaluations of school-based interventions on delaying sexual debut suggest mixed findings.[40 42–44] Within high-income countries, the majority of school-based interventions have shown effectiveness in delaying sexual debut.[42–44] Similarly, a systematic review and meta-analysis on school based sex education and HIV prevention in low-income and middle-income countries globally found that students who received the interventions were less likely to initiate sexual activity.[45] In a cluster randomised controlled trial (RCT) assessing the effects of teacher-led school HIV prevention programmes on adolescent sexual risk behaviour in Dar es Salaam, Tanzania and Cape town and Mankweng, South Africa students in Tanzania reported delay in initiating sexual activity during the study while there was no effect of the intervention among students in South Africa.[46] Other studies found no significant effects of school-based interventions on delaying sexual debut[47 48] while others found significant effects[49 50] found significant effects. These conflicting findings suggests a need for a comprehensive review of school-based interventions to assess their effectiveness on delaying sexual debut.

Taken together, these factors underscore the need to synthesise existing studies, and explore the linkages between school-based interventions and early sexual debut in an attempt to inform future adolescent health policies and programing. This systematic review will provide a critical synthesis of existing literature on school-based interventions aimed at delaying early sexual debut among adolescents in SSA, to inform programmes, policy and research. The objective of the review is to evaluate the effects of school-based interventions on delaying sexual debut among adolescents in SSA.

## Methods and design

This systematic review and meta-analysis will be developed in accordance with the Preferred Reporting Items for Systematic Reviews and Meta-Analyses Protocols.[51] Criteria for considering studies will include study population, type of interventions, type of outcome measures, and type of studies, and study setting, as described in the sections that follow. Important amendments made to this protocol will be documented and published alongside the results of the systematic review.

## Study population

The study targets adolescent students aged 10–19 years in primary or secondary levels of education or their equivalents who participated in a school-based intervention to delay sexual debut. Studies with students younger than 10 years and/or older than 19 years, as maybe in some settings, will be included if the majority of participants (ie, above 50%) are aged between 10 and 19 years. Studies that do not include students aged 10–19 years will be excluded from the review.

## Types of interventions

All interventions with a school-based component irrespective of intervention content and instruction mode will be included as long as they assessed sexual debut as a primary or secondary outcome. This includes interventions that are school-based only and delivered in primary and secondary schools, and those that have multiple components, one of which must be a school-based component while the other components are delivered elsewhere (eg, healthcare facilities). The intervention must have reported on sexual debut as a primary or secondary outcome. Interventions that target students outside the school setting will be excluded.

## Types of outcome

The primary outcome of interest is delayed sexual debut, defined as postponement of sexual intercourse among participants who had not engaged in sexual intercourse prior to the school-based intervention. While secondary outcomes shown in table 1 will be considered as a first step, other SRH outcomes reported by at least two studies will also be considered. Such may include contraceptive use, pregnancy and history of sexually transmitted infections (STIs).

**Table 1** Secondary outcomes

| Secondary outcomes | Definitions |
|---|---|
| Intention to delay sex | Planning to wait to have sexual intercourse until older |
| Lifetime sexual activity | Ever engaged in sexual intercourse |
| Sexually active | Having engaged in sexual intercourse in the last 30 days |
| Current sexual activity | Having engaged in sexual intercourse in the last 6 months |
| Sexual health knowledge | Knowledge of key SRH topics and issues |
| Sexual attitude | Attitudes towards sexuality and sexual behaviour |
| Self-efficacy for safe sex practices | Confidence to say no to unsafe sex practices |
| Consistent condom use | Condom use at every sexual intercourse |
| No of sex partners | No of sex partners (regular and casual) in a specified period of time |

SRH, sexual and reproductive health.

## Types of studies
The review will include all studies focused on delaying sexual debut, regardless of the study design employed. Meta-analytical approaches that take into consideration different study designs will be used in the analysis, and where meta-analysis is not possible, other methods of analysing the effect measures outlined by McKenzie and Brennan[52] will be employed . We will also compare the effectiveness of the interventions by study design. Eligible studies will have been published in peer-reviewed journals between January 2009 and December 2020. Only studies published in English will be included.

## Study setting
Only studies conducted in SSA will be included. Given the population of interest, the review will focus on interventions implemented in primary and secondary schools or their equivalents. The review focuses on studies published since 2009, the period after the global momentum on the need for comprehensive sexuality education began. In 2008, Latin America and the Caribbean signed the Preventing through Education Declaration for the delivery of sexuality education and health services.[53] In 2013, there was a Ministerial Commitment on comprehensive sexuality education (CSE) and SRH services for adolescents and young people in 20 countries across Eastern and Southern Africa.[54]

## Search strategy
Our search strategy will involve three methods:
► Electronic searches: four researchers will search five electronic databases including PubMed, Scopus, Science Direct, Web of Science and EBSCO (PsycINFO, Global Health, CINAHL) for published peer-reviewed journal articles. Cochrane library, National Institute of Health and Turning Research into Practice databases will be searched for ongoing studies that are yet to be published.
► Handsearches: an iterative process to obtain additional studies not yet retrieved with our initial online database using the reference list of retrieved articles

► Contacting authors and experts: where published data are not sufficient, authors will be contacted for additional information

While the exact search terms will vary by database, the four search components included will be[1] adolescents[2] sexual debut[3] school-based interventions[4] and SSA. Search terms will be adapted for each bibliographic database in combination with database-specific filters. Boolean operators 'OR', 'NOT' and 'AND' will be used to maximise or narrow the specificity for the search. Wildcards will be used to search for variations or alternate spellings of key concepts. Below, and in the supplementary document(online supplemental file 1), we present a search strategy to be used in PubMed below:

1. (intervention[Title/Abstract]) OR (program*[Title/Abstract])
2. (school[Title/Abstract]) OR (institution[Title/Abstract]) OR (academic[Title/Abstract]) OR (education[Title/Abstract])
3. (sexual debut[Title/Abstract]) OR (sexual initiation[Title/Abstract]) OR (sexual delay[Title/Abstract]) OR (sexual activity [Title/Abstract])
4. (adolesce*[Title/Abstract]) OR ("young people"[Title/Abstract]) OR (youth[Title/Abstract]) OR (teenage*[Title/Abstract]) OR (learner[Title/Abstract]) OR (children[Title/Abstract])
5. (Africa south of the Sahara [MeSH Terms]) OR (Africa [MeSH Terms])
6. ("2009/01/01"[Date - Publication]: "2020/12/31"[Date - Publication])
7. #1 AND #2 AND #3 AND #4 AND #5 AND #6 ("intervention"[Title/Abstract] OR "program*"[Title/Abstract]) AND ("school"[Title/Abstract] OR "institution"[Title/Abstract] OR "academic"[Title/Abstract] OR "education"[Title/Abstract]) AND ("sexual debut"[Title/Abstract] OR "sexual initiation"[Title/Abstract] OR "sexual delay"[Title/Abstract] OR "sexual activity"[Title/Abstract]) AND ("adolesce*"[Title/Abstract] OR "young people"[Title/Abstract] OR "youth"[Title/Abstract] OR "teenage*"[Title/

Abstract] OR "learner"[Title/Abstract] OR "children"[Title/Abstract]) AND ("africa south of the sahara"[MeSH Terms] OR "africa"[MeSH Terms])[14]AND 2009/01/01:2020/12/31[Date - Publication]

## DATA COLLECTION

### Selection of studies

Four reviewers, paired, will independently screen retrieved titles and abstracts of potential articles to determine their eligibility for inclusion in the review. Rayyan, a web application tool will be used to manage the screening process.[55] Full article texts of included studies will then be obtained and reviewed to ascertain eligibility. Any disagreements during title and abstract review or during the full text review will be resolved by consensus. A third reviewer will be involved if a consensus is not reached by the two pairs of reviewers.

### Data extraction and management

Two reviewers will use a standardised data extraction form to independently extract data on background or process-oriented information from each included study to provide a basis for data charting, themes and variables for use in answering the research question. Reviewers will pilot the data extraction form with a sample of included papers and amendments will be made as necessary. From each relevant study, data will be extracted on the following domains:

► General information on the study: authors, date of publication, publication type, country and funding source.
► Study characteristics: study setting, location, study design, sampling frame and sampling methods, and year(s) of study implementation.
► Participant characteristics: age, gender, number of participants, participants lost to follow-up, length of follow-up.
► Intervention: detailed description of the intervention, composition of intervention and control groups.

Outcomes: for primary outcome (delay in sexual debut), data on the number of participants experiencing delayed sexual debut on both intervention versus control arms will be extracted. If a summary estimate (eg, risk ration) is reported instead of raw numbers, these will be extracted. A similar approach will be used for secondary outcomes. Where more than one article described the same intervention, data will be extracted from all papers. The eligible studies will be exported into the Comprehensive Meta-Analysis[56] software.

### Risk of bias assessment

Two independent reviewers will assess the methodological quality of studies depending on study design. For RCTs, the reviewers will judge each quality domain based on the following three-point scale as suggested in Cochrane Handbook for Systematic Reviews of Interventions[57]: Yes (low risk of bias: plausible bias unlikely to seriously alter the results if all criteria were met); No (high risk of bias: plausible bias

that seriously weakens confidence in the results if one or more criteria were not met) and Unclear (plausible bias that raises some doubt about the results if one or more criteria were assessed as unclear. The following items in the risk of bias assessment for RCTs will be included:

► Sequence generation (whether allocation sequence was adequately generated).
► Allocation concealment (whether allocation was adequately concealed).
► Masking/blinding (whether knowledge of the allocated intervention was adequately prevented during the study, that is, whether participants, personnel, outcome assessors and/or data analysts are blinded).
► Incomplete outcome reporting (whether incomplete outcome data was adequately addressed).
► Selective reporting (whether reports of the study were free of selective outcome reporting).
► Other sources of data (eg, whether reports of the study included sample size computation, alpha error, etc.).

For non-RCT studies, the following items will be included in the risk of bias assessment as suggested by Viswanathan *et al.*[58]

► Selection bias: do the inclusion/exclusion criteria vary across the comparison groups (for multiple-arm studies) or within groups (for single-arm/cross-sectional studies)? Performance bias—does the study fail to account for important variations in the execution of the study from the proposed protocol?
► Detection bias: was the assessor not blinded to the outcome, exposure, or intervention status of the participants? Were valid and reliable measures not used or not implemented consistently across all study participants to assess inclusion/exclusion criteria, intervention/exposure outcomes, participant benefits and harms?
► Attrition bias: was the length of follow-up different across study groups? In cases of missing data was the impact not assessed (eg, through sensitivity analysis or other adjustment method)?
► Selective outcome reporting: are any important primary outcomes missing from the results?
► Overall assessment: are the results believable taking study limitations into consideration?

Discussions and consensus will be used to crosscheck the extracted information and to resolve disagreements. The level of risk of bias in each of these domains will be presented separately for each study in tables in the final review publication.

### Risk of bias in individual studies

Methodological quality of each individual article will be appraised using a checklist adapted from Critical Appraisals Skills Programme. Two authors will independently appraise each article.

### Measures of intervention effect

Dichotomous outcomes such as sexual debut (initiated vs not initiated sexual activity during the study period)

will be summarised as risk ratios or OR with 95% CIs for each study. Continuous outcomes such as number of sex partners will be summarised as (unstandardised) mean differences and SEs. In cases where sexual debut may be reported as a median age at sexual debut, published procedures for transforming median to mean and SD to facilitate computation of relevant statistics will be used.[59] If enough information is not provided to calculate an effect size, study authors will be contacted for clarification or to provide additional statistics. If the authors do not provide this information after 1 month, this study will be removed from the analysis but this effort will be reported.

If results of a repeated measure analysis are reported, authors will need to provide the correlation between pre-post measurements or provide enough information to calculate the correlation between measurements. If these statistics are not available, either in publication or after request, and the study was a controlled design, an effect size will be generated using postintervention statistics provided groups are similar at baseline with respect to the outcome of interest and other relevant covariates.

### Unit of analysis

Data will be extracted from each included study (unit of analysis) as follows. For dichotomous variables, number of participants in the intervention group and the number of participants in the standard/control group will be used. For continuous variables, the mean, SD, and the number of participants in the intervention and control groups will be used. For studies with multiple intervention groups, each pair-wise comparison will be included separately. Moreover, for dichotomous outcomes, we will divide both the number of events and the total number of participants. For continuous outcomes, means and SD will not be changed but will be divided by the total number of participants.

### Assessment of heterogeneity

Heterogeneity among studies and between subgroups will be assessed using a $\chi^2$ test with a significance level at p<0.10. The degree of heterogeneity among studies and between subgroups will be assessed using the $I^2$ statistic using the following guide: $I^2$=0% to 40% as heterogeneity that might not be important, $I^2$=30% to 60% as moderate heterogeneity, $I^2$=50% to 90% as substantial heterogeneity and $I^2$=75% to 100% as considerable heterogeneity[60]

### Assessment of reporting bias

Publication bias will be assessed if at least 10 studies are included in the review. A funnel plot will be used to assess the magnitude of reporting bias as per Cochrane guidelines.

### DATA SYNTHESIS

Data will be pooled using the random effects model according to Chapter 9 of the *Cochrane Handbook for Systematic Reviews of Interventions*.[60] Dichotomous outcomes'

data (eg, number of participants who experience sexual debut) will be reported as risk ratios and continuous data (eg, number of sexual partners) as mean differences. Depending on the different types of interventions identified, such as sexuality education, economic interventions or a combination of different approaches, we will conduct pooled effects for each type of interventions to examine which are more effective in delaying sexual debut. Additionally, and where possible, we will identify and consider conceptual groupings such RCTs and non-RCT studies while conducting the analysis. All analyses will be done at 95% CIs. Data will be analysed using the statistical software Comprehensive Meta-Analysis.[56] In studies where the effects of clustering have not been taken into account, SD will be adjusted for the design effect, using intra-class coefficients, if they are provided in the study reports, or alternatively using external estimates obtained from similar studies. Stratifications will be done by school level (primary or secondary (or age, if applicable), grade level, instructor (eg, teacher or peer), intervention type (eg, behavioural change communication, study setting).

Other acceptable data synthesis methods such as those descried by McKenzie and Brennan[52] will be considered where meta-analysis is not possible. Such methods include summarising effect estimates in case estimates of intervention effects are available but the variances of the effects are not reported or are not correct.[52] Another method described by McKenzie and Brennan[52] is combining P values if no other information is available; differing statistical tests and outcomes across studies or non-parametric test results reported. Lastly, vote counting based on the direction of the effect can be used when only the direction of effect is reported or there is no consistent effect measure or data reported across studies.[52]

Findings will be organised in tables and charts, while presenting a description of the themes and concepts found in literature reflecting the review objectives. A summary narrative that syntheses the information across tables and charts will be developed, critically highlighting the advances and gaps in research, with a focus to draw implications for future research.

**Contributors** BWM, KJ, EKI and JO contributed to the conceptualisation, and development of the protocol. BWM and KJ provided the initial draft of the protocol. HW and CWK provided a critical review of the protocol. All authors read and approved the final draft of the protocol.

**Funding** Authors' time to develop this protocol was supported by the International Development Research Centre (grant number 108676-002) and the Swedish International Development Cooperation Agency (grant number 12103).

**Competing interests** None declared.

**Patient consent for publication** Not required.

**Provenance and peer review** Not commissioned; externally peer reviewed.

of the translations (including but not limited to local regulations, clinical guidelines, terminology, drug names and drug dosages), and is not responsible for any error and/or omissions arising from translation and adaptation or otherwise.

**ORCID iDs**
Beatrice W Maina http://orcid.org/0000-0001-6205-3296
Kenneth Juma http://orcid.org/0000-0001-7742-9954

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
