## [Reviewer comments · BMJ Open]

ARTICLE DETAILS

TITLE (PROVISIONAL)	Effectiveness of school-based interventions in delaying sexual debut among adolescents in sub-Saharan Africa: A protocol for a systematic review and meta-analysis
AUTHORS	Maina, Beatrice; Juma, Kenneth; Igonya, Emmy; Osindo, Jane; Wao, Hesborn; Kabiru, Caroline

VERSION 1 – REVIEW

REVIEWER	Peterson, Amy ETR Scotts Valley
REVIEW RETURNED	18-Nov-2020

GENERAL COMMENTS	The authors describe a protocol for a systematic review and meta-analysis for assessing the effects of school-based interventions in delaying sexual debut among adolescents in sub-Saharan Africa (SSA). The authors provide a rationale that school is an important setting for adolescent health and that there are existing school-based interventions in SSA that have been evaluated, warranting a review on the extent of the effects of these interventions on sexual debut. Overall, the protocol provides a clear strategy for searching, screening, extracting, appraising and synthesizing studies. There are a few areas where the protocol can be strengthened, particularly around interventions of interest and approach to synthesis. I would suggest a revision to further clarify the types of interventions you plan to include and how this will shape decisions in your data collection and analysis. In the introduction, you mention both sex education as well as economic interventions (e.g. school fees), though specific examples of interventions tend to focus on sex education. Your methods allow for “all interventions with a school-based component irrespective of intervention content or instruction mode will be considered” -- your use of the word ‘considered’ here is confusing – will all studies that meet this criteria be included? And if not, which studies will be? Given the wide range of studies you may get as a result of this criteria, will you plan to account for different types of interventions in the synthesis stage, i.e. will you conduct pooled effects for different types of interventions (i.e. sex education vs. economic interventions)? Another suggested revision is to further describe your approach to synthesis. The methods describe a meta-analytical approach by outcome but does not consider other conceptual groupings that might arise from the included studies. For example, there is no indication that studies with more rigorous study designs (i.e. RCTs) will be pooled separately from one-arm, noncomparative studies), even though this may make interpreting pooled effects more difficult. Please consider how you will go about assessing
---

	studies for appropriate conceptual groupings such as intervention type, study design and quality, heterogeneity, etc. These groupings may also be supportive of any narrative synthesis you conduct. Please also consider other methods if meta-analysis is not possible (i.e. Chapter 12 in Cochrane Handbook and Synthesis Without Meta-Analysis guidelines). Additional comments  -The SSA setting seems to be an important justification for conducting this review. Can you provide more detail in your introduction on whether similar reviews have focused on the SSA setting or included SSA specifically? -Sexual debut is clearly your primary outcome. For secondary outcomes, can you provide a justification for those listed as opposed to others (e.g. contraception, pregnancy, STDs, etc)? -Unless data collection has already been conducted, consider including studies from 2020. Please also describe the rationale for the 2009 start date. -I do not understand line 187-188 – it looks like it might be missing words -Can you provide a rationale for including noncomparative studies? Is there a lack of RCT or quasi-experimental studies for sexual health interventions in SSA? -Random effects is mentioned in the abstract but not discussed in the protocol -It is stated that ethics is not required but does not indicate which body determined this. -The term 'considered' is used both in the abstract and in several places in the protocol – please modify to 'include' or provide more information about which studies will be included
--	--

REVIEWER	Marino, Jennifer L. Univ Melbourne, Obstetrics and Gynaecology, Royal Women's Hospital
REVIEW RETURNED	22-Dec-2020

GENERAL COMMENTS	Thank you for the opportunity to review this protocol, which describes an ambitious and important systematic review and meta-analysis of the effectiveness of school-based interventions in delaying first sexual intercourse in adolescents in sub-Saharan Africa.  1. I note that this review has not been registered with PROSPERO (or equivalent register). Is there a plan to register it? 2. I am not entirely clear how the screening method works – will the two teams split the list, and each reviewer within a team double reviews each title/abstract on that team's half, with consultation with the other team if there is dissent? Or does each team review all the references, discuss any tricky ones, and then the two teams compare? I am unclear where the third reviewer comes when all the references have already been double (in some cases quadruple?) screened. 3. Please provide the citation for the CMA software package (first mention, l.203, p.10) as you have for Rayyan. If there is not a formal citation, usually company name, country and version number or year of most recent release will suffice. 4. Please provide the reference(s) for the published methods to convert median to mean and standard deviation (ll. 250-2, p. 12).
--

VERSION 1 – AUTHOR RESPONSE

Reviewer 1:

1. The authors describe a protocol for a systematic review and meta-analysis for assessing the effects of school-based interventions in delaying sexual debut among adolescents in sub-Saharan Africa (SSA). The authors provide a rationale that school is an important setting for adolescent health and that there are existing school-based interventions in SSA that have been evaluated, warranting a review on the extent of the effects of these interventions on sexual debut. Overall, the protocol provides a clear strategy for searching, screening, extracting, appraising and synthesizing studies.

Response: Thank you

2. There are a few areas where the protocol can be strengthened, particularly around interventions of interest and approach to synthesis.
I would suggest a revision to further clarify the types of interventions you plan to include and how this will shape decisions in your data collection and analysis. In the introduction, you mention both sex education as well as economic interventions (e.g. school fees), though specific examples of interventions tend to focus on sex education

Response: We will include all interventions aimed to delay sexual debut. In addition to interventions focusing on sexuality educations, we recognize there are other interventions, either in combination with sexuality education or implemented as a stand-alone intervention, such as those offering school support to vulnerable and most-at-risk adolescents. As such, we will consider all the interventions implemented in schools to delay sexual debut. (see lines 133 – 139)

3. Your methods allow for “all interventions with a school-based component irrespective of intervention content or instruction mode will be considered” -- your use of the word ‘considered’ here is confusing – will all studies that meet this criteria be included? And if not, which studies will be? Given the wide range of studies you may get as a result of this criteria, will you plan to account for different types of interventions in the synthesis stage, i.e. will you conduct pooled effects for different types of interventions (i.e. sex education vs. economic interventions)?

Response: We will include all studies that meet this criterion if they measured sexual debut as an outcome. We have replaced considered with included throughout the text. We will account for the different types of interventions and compare them to examine which interventions are more effective in delaying sex debut as indicted in line 137 - 139

4. Another suggested revision is to further describe your approach to synthesis. The methods describe a meta-analytical approach by outcome but does not consider other conceptual groupings that might arise from the included studies. For example, there is no indication that studies with more rigorous study designs (i.e. RCTs) will be pooled separately from one-arm, noncomparative studies), even though this may make interpreting pooled effects more difficult. Please consider how you will go about assessing studies for appropriate conceptual groupings such as intervention type, study design and quality, heterogeneity, etc. These groupings may also be supportive of any narrative synthesis you conduct.

Response: We have added a section to this effect. As earlier indicated, we will account for the different types of interventions by conducting pooled effects for different types of interventions (RCTs, one-arm, non-comparative studies) and compare them to examine which interventions are more effective in delaying sex debut. See lines 286 - 290

5. Please also consider other methods if meta-analysis is not possible (i.e. Chapter 12 in Cochrane Handbook and Synthesis Without Meta-Analysis guidelines).

Response: Thank you. We have added a section on other methods that we will consider in case meta-analysis is not possible. See lines 298 - 305

Additional comments

6. The SSA setting seems to be an important justification for conducting this review. Can you provide more detail in your introduction on whether similar reviews have focused on the SSA setting or included SSA specifically?

Response: To the best of our knowledge, there is no other review that has focused on the effectiveness of school-based interventions on delaying sexual debut. A majority of existing reviews in SSA or low-income settings have focused on STI/HIV preventions and have mainly targeted interventions on sex education. For instance, a review by Fonner et al (2014) focused on “School based sex education and HIV prevention in low-and middle-income countries: a systematic review and meta-analysis”

7. Sexual debut is clearly your primary outcome. For secondary outcomes, can you provide a justification for those listed as opposed to others (e.g. contraception, pregnancy, STDs, etc)?

Response: We acknowledge our list of secondary outcomes is not exhaustive. We have added a sentence to show that other secondary outcomes reported by at least two studies will be considered. See lines 130 - 131

8. Unless data collection has already been conducted, consider including studies from 2020. Please also describe the rationale for the 2009 start date.

Response: On lines 145 - 150, we have provided a justification on for 2009 as a start date. We will include studies conducted in 2020 as well.

9. I do not understand line 187-188 – it looks like it might be missing words

Response: The sentence was misplaced and we have deleted it.

10. Can you provide a rationale for including non-comparative studies? Is there a lack of RCT or quasi-experimental studies for sexual health interventions in SSA?

Response: Our rationale for including non-comparative studies is because these studies are often used in the evaluation of healthcare and public health interventions in cases where randomization is impossible

11. Random effects is mentioned in the abstract but not discussed in the protocol

Response: This is discussed in in the protocol. See from line 286

12. It is stated that ethics is not required but does not indicate which body determined this.

Response: We are not directly involved with study participants and so not collecting personal, sensitive or confidential information. Our data is publicly available as evidence and thus, we presume, no need to seek institutional ethical approval.

13. The term ‘considered’ is used both in the abstract and in several places in the protocol – please modify to ‘include’ or provide more information about which studies will be included

Response: We have replaced considered with included throughout the text.

Reviewer: 2

Comments to the Author:

Thank you for the opportunity to review this protocol, which describes an ambitious and important systematic review and meta-analysis of the effectiveness of school-based interventions in delaying first sexual intercourse in adolescents in sub-Saharan Africa.

1. I note that this review has not been registered with PROSPERO (or equivalent register). Is there a plan to register it?

Response: Yes, we will register the review with PROSPERO

2. I am not entirely clear how the screening method works – will the two teams split the list, and each reviewer within a team double reviews each title/abstract on that team's half, with consultation with the other team if there is dissent? Or does each team review all the references, discuss any tricky ones, and then the two teams compare? I am unclear where the third reviewer comes when all the references have already been double (in some cases quadruple?) screened.

Response: We will use Rayyan—a web application—to screen articles. The application allows you to set the number of reviewers needed to screen an article and will show articles that have not been screened by the maximum number required. Articles are assigned randomly and we will review them blindly. Once an article is reviewed, the reviewer indicates include or exclude. Articles are included or excluded if both reviewers are in agreement. If there is a disagreement, a third reviewer breaks the tie. Some articles might require a discussion with the whole team to determine if they are to be included or not.

3. Please provide the citation for the CMA software package (first mention, l.203, p.10) as you have for Rayyan. If there is not a formal citation, usually company name, country and version number or year of most recent release will suffice.

Response: A citation has been provided

4. Please provide the reference(s) for the published methods to convert median to mean and standard deviation (ll. 250-2, p. 12)

Response: A reference has been added

VERSION 2 – REVIEW

REVIEWER	Peterson, Amy ETR Scotts Valley
REVIEW RETURNED	29-Mar-2021

GENERAL COMMENTS	Thank you for the chance to review the revisions to the protocol. The author have sufficiently responded to all raised concerns.
--

REVIEWER	Marino, Jennifer L. Univ Melbourne, Obstetrics and Gynaecology, Royal Women's Hospital
REVIEW RETURNED	18-Mar-2021

GENERAL COMMENTS	Thank you for the opportunity to review this protocol again. My thanks to the authors for their careful attention to my suggestions, which I feel have been sufficiently addressed.
---